# Peer review of "Crowdsourcing in medical research: concepts and applications"

_PeerJ, doi:10.7717/peerj.6762_

## Round 0.1 · original submission · Major Revisions

· Academic Editor

Major Revisions

Making the required revisions for one reviewer will greatly satisfy the other reviewer, especially where they overlap on expected scope, breadth, and requisite definition. With that said, and since another reviewer rejected the manuscript early on, given the timeframe and an apology for the holiday interference, I am going forward with the decision of major revisions with expectation that this manuscript will greatly improve from all requested attention to detail and specifics.

Reviewer 1 ·

Basic reporting

The paper is written in clear, professional style. The references are plentiful and relevant, and the topic is current, as interest in crowdsourced approaches to biomedical research is surging.

However, some of the statements made are taken as facts and not adequately challenged. For instance, in lines 140-141, the authors claim that "Empirical evidence from humans suggests that there is a convergent collective intelligence factor that explains a group’s performance on several tasks." Though the authors claim rests on a publication, it is debatable. Convergence is typically the result of an interaction between the participants. If there is no interaction -- and crowdsourcing does not require it -- convergence need not happen.

The authors also claim that a feature of crowdsourced models is to share the solution with the public (lines 156-157). This applies to some crowdsourced models, but is by no means a requirement. For instance, Innocentive, which is a platform that taps the crowd to solve difficult scientific challenges, does not share its solutions. Likewise, some public-private partnerships, such as the Medicines for Malaria Venture, use a crowdsourcing approach (open calls on the internet) to identify novel ways to possibly tackle malaria. Ideas are submitted independently, and are evaluated by a review board to determine which will be pursued, but they are not shared with the public. In fact, the output of the crowdsourcing process in this particular case is not even a solution, but an input into the research process. Yet, that does not make it less valuable.

Finally, the authors spend much time discussing the application of crowdsourcing to community-based research. This is just a subset of crowdsourcing applications, and whatever conclusions they reach in that setting should not be generalized to the entire field. Designing therapeutic programs that are embraced a patient community is quite different from creating a low-cost process to synthesize a new molecule. Crowdsourcing models can be used for both, but they will likely look quite different.

The authors make further unjustified generalizations in the next section (lines 230-275), when they claim that "Issues that are more amenable to crowdsourcing approaches include specific behavioral or social issues (e.g., changing condom use behaviors), topics that have champions and are timely, topics with robust ally networks, and topics where the public has a responsibility to be engaged.". This statement is not properly documented, and takes a narrow view of crowdsourcing. At its core, crowdsourcing consists in tapping a broad range of expertise and ideas to solve problems. In biomedical research, it has proven useful in solving problems ranging from tough scientific challenges to designing community-based disease interventions. But there is no evidence to say that it works better in the latter case. In fact, both are hardly comparable.

The section about the "Critiques of Crowdsourcing" dwells too much on criticism that should be quickly disposed of. It is obvious that any problem-solving approach that taps the crowd is likely to generate a certain volume of low-quality input -- the so-called "madness of groups". However, this is irrelevant. What makes the method valuable is not the low end of the spectrum but its high end. The latter is the only thing that matters. In the same section, the "Boaty McBoatface" should be explained.

In the last section about "Crowdsourcing Applications in Health", the authors mention applications such as "develop study names, logos" and "images, videos, slogans" while overlooking the entire field of drug R&D. Yet, one would expect the latter to be of significantly greater value than logos and slogans. There is an imbalance here that is troublesome.

Experimental design

This is a literature review, and does not involve a study design.

Validity of the findings

It seems that the authors set out to lay the foundation of a "crowdsourcing theory" (lines 60-65), and that perhaps is the problem. Crowdsourcing is a problem-solving approach that works. Its effectiveness is supported by a large body of evidence, and it is not clear why it needs a theory. One may legitimately wonder why it works and if it can be improved, but these questions hinge on what drives individual and collective creativity, topics that are still poorly understood. One may recommend additional research to address them, but this is not what this paper does. It tries instead to build a theory upon a foundation that is too fragmented to support it and, in doing so, disregards important applications of crowdsourcing (e.g., drug R&D), and extrapolates findings from another subset (community-based disease interventions) that do not apply to the entire field.

·

Basic reporting

The article is written in clear, unambiguous, professional English and is structured professionally. The article is mostly adequately referenced. However, it would seem that references are needed on lines 62, 232, and 262. There are a few minor typos (lines 256 and 285).

The review is broad and cross-disciplinary, but to the extent that it loses the ability to adequately summarize the literature. The terms "medical research", "medicine", and "health" are used interchangeably. Yet "health" is much wider in scope than "medical research". This lack of focus results in a jumble of findings where examples are given on extremely heterogeneous aspects of health, including the use of crowdsourcing for public health interventions as well as for basic research of pathogens. This does not allow for a clear summary of the findings from the literature.

The introduction introduces the subject of crowdsourcing, with an unusual definition. The cited definition focuses on a group of individuals proposing solutions to a problem and sharing these solutions. However, most definitions of crowdsourcing focus on dividing work amongst participants, gathered through the internet or other public platforms. The authors state (lines 81-83) that "this definition is more relevant to medical applications and provides a clearer outline of the essential aspects of crowdsourcing - group participation in problem solving and sharing solutions widely." Yet the authors do not explain why dividing work amongst a group of volunteers is irrelevant to medical applications. Whereas some of the examples mentioned later in the article are performing exactly this type of crowdsourcing. The second aspect of the definition (publicly sharing) is also not always the case. Many crowdsourcing applications actually utilize intellectual property rights and are not restricted to create commons licenses.

It appears that the authors’ motivation for the article is to create a theoretical foundation for crowdsourcing for medicine (lines 56-58). However, this motivation does not fit with a literature review, where the reader expects a summary of the existing evidence/literature.

Experimental design

The survey methodology is insufficiently described. The sources are given (PubMed and Google Scholar) but no information is given regarding the search terms, the search dates, the inclusion and exclusion criteria, how many articles were returned, and how many were excluded. Additionally the parts of the theory (i.e., collective intelligence, open source model, etc.) are not described so that the reader is unsure if this is the authors' own theory or based upon an external reference.

Validity of the findings

As mentioned before, since the literature review does not refine the research question, the results are a mix of different crowdsourcing findings applied to widely divergent tasks. Some examples do not actually relate to crowdsourcing, for example, the forwarding of an email (lines 258-262). The authors do narrow down the potential useful health-related areas of crowdsourcing, focusing mostly on behavioral and social science-related health challenges. In retrospect, the literature review would have been significantly better if the authors had started the review here and provided the reader with a better summary of the evidence as it relates to the strengths and weaknesses of crowdsourcing of behavioral and social science-related health questions/interventions. I hope that any revisions are directed to refining the literature review in this way.

The authors appear to not take the criticisms of crowdsourcing seriously, giving one reference for a potential mitigation strategy to avoid "mad" groups (line 297) and yet this reference article does not relate to crowdsourcing but a moderated, specific intervention. The example of scepticism regarding vaccines seems like an obvious example where "mad" groups have had a significant impact.

Reviewer 3 ·

Basic reporting

Needs:

The article would benefit from a clear and consistent definition of crowdsourcing, and it's relationship to open innovation. This is no easy task, as both academics and practitioners have not yet agreed on a common nomenclature. That being said, synthesizing the concept of open innovation (Chesbrough) would be helpful.

In terms of consistency, the introduction often feels at odds with subsequent sections. For example, the rich history of challenges (e.g. Longitude Prize) is noted in detail in a later section, and yet the introduction (line 54) suggests that 54. “the concept has spurred diverse health programs, including challenge contests….”

At a minimum, the authors are advised to acknowledge how definitions vary, and perhaps include a table of the larger vocabulary employed by practitioners.

A second area for clarification is the definition of "medicine." At times, the paper suggests that the definition is broad. It would be wise to define "medicine" upfront.

In general, I found the article somewhat hard to follow. The majority is spent describing theories, and the conclusion comes quickly at the end without much discussion of other types of crowdsourcing activities, and why they may or may not be successful/appropriate.

Experimental design

As a practitioner, blindspot in the literature, and therefore, a literature review, is that a crowdsourcing program's design has a significant effect on outcome. Rarely is the design taken into account - or even known - by outsiders.

It is widely known among practitioners that poorly-designed crowdsourcing programs yield negative results. A few common mistakes that affect outcome:

- Lack of a clear problem statement or call-to-action
- Misalignment of incentives and intellectual property stance
- Lack of funds or expertise to effectively engage the crowd

Validity of the findings

The conclusion in the section "Crowdsource-able Challenges in Health" feels abrupt I am not seeing the connection between these types of challenges being more appropriate/useful. We have seen crowdsourcing programs effectively tackle a broad cross-section of topics and solutions across health and healthcare. Pillpack, for example, is the product of an MIT hackathon. It was acquired by Amazon for over $1 billion.

Perhaps this is the challenge with literature review vs. study of crowdsourcing outcomes themselves. (see above).

Additional comments

I commend you for taking up this subject. Following the 20th century, where most problem solving was considered proprietary and confidential, we are indeed entering into a new era where collaboration is the new competitive edge.

Cleaning up definitions and nomenclature will greatly improve the article. Specifically, ensuring that the introduction is concise - there are many places where the intro felt off, but later in the article a more thoughtful statement is made.

In terms of the findings, I am at a loss as it's not spelled out in the piece how you arrived at what "good" or "useful" is.

Annotated reviews are not available for download in order to protect the identity of reviewers who chose to remain anonymous.

---

## Round 0.2 · Minor Revisions

· Academic Editor

Minor Revisions

Please consider the latest peer review comments and make changes to your manuscript as appropriate. My main concern with the manuscript stems from the one-day visit to PubMed and Google Scholar that leaves off visits to ResearchGate and Academia.com, the latter of which may have archived works not on the former two. While that is water under the bridge in a sense, it could still behoove the authors to at least note an updated attempt to relay their eventual access and consideration to those sites seeing that there is relevant data there, and so as to not have the sites jump out as neglected for a literature review, despite obvious probability of redundancy between former and latter mentions.

·

Basic reporting

The article is clear with professional English used throughout. There is a minor typo on line 91. The article is professionally structured.

Experimental design

The methods section is much improved in this version.

Validity of the findings

Please check the definition of open innovation (line 55-56). Chesborough's definition is that companies use external sources, but it does not include any reference to transparency or making them publicly available. I believe that you are using an alternative defintiion here.

On lines 92-97, it would be good to state that other incentives exist for contributions. For example, Innocentive gives monetary compensation.

OSDD (lines 187-189) is no longer operational. https://www.bbc.com/news/business-42188808

Lines (294-6) state that "Crowdsourcing approaches have been used to develop antibiotics, lupus drugs, and anti-malarials." I believe that it should state that "Crowdsourcing approaches have been used to assist in the discovery and development of...".

Lines (300-302) state that "Crowdsourcing could accelerate several stages of drug development, including screening, pre-clinical trials, and human clinical trials." Although the additional information added later about human clinical trials is about study names and logos. These aspects do not greatly delay clinical trials. I would suggest that human clinical trials are removed here since there are many serious problems with the idea of crowdsourcing these trials, for example, maintaining a credible randomized controlled trial to eliminate bias.

Additional comments

Thank you for your attention to the reviewers' comments. This revised article is significantly strengthened.

---

## Round 0.3 · accepted · Accept

· Academic Editor

Accept

The revisions has adjusted concerns of the last peer reviewer. This decision is made contingent on PeerJ final approval to properly style the citations in this document and its references list or to advise otherwise.